# Identifying Coarse-grained Independent Causal Mechanisms with Self-supervision

## Abstract

Current approaches for learning disentangled representations assume that independent latent variables generate the data through a single data generation process. In contrast, this manuscript considers independent causal mechanisms (ICM), which, unlike disentangled representations, directly model multiple data generation processes (mechanisms) in a coarse granularity. In this work, we aim to learn a model that disentangles each mechanism and approximates the ground-truth mechanisms from observational data. We outline sufficient conditions under which the mechanisms can be learned using a single self-supervised generative model with an unconventional mixture prior, simplifying previous methods. Moreover, we prove the identifiability of our model w.r.t. the mechanisms in the self-supervised scenario. We compare our approach to disentangled representations on various downstream tasks, showing that our approach is more robust to intervention, covariant shift, and noise due to the disentanglement between the data generation processes.

## 1 Introduction

The past decade witnessed the great success of machine learning (ML) algorithms, which achieve record-breaking performance in various tasks. However, most of the successes are based on discovering statistical regularities that are encoded in the data, instead of causal structure. As a consequence, standard ML model performance may decrease significantly under minor changes to the data, such as color changes that are irrelevant for the task, but which affect the statistical associations. On the other hand, human intelligence is more robust against such changes (Szegedy et al., 2013). For example, if a baby learns to recognize a digit, the baby can recognize the digit regardless of color, brightness, or even some style changes. Arguably, it is because human intelligence relies on *causal mechanisms* (Schölkopf et al., 2012; Peters et al., 2017) which make sense beyond a particular entailed data distribution (Parascandolo et al., 2018). The independent causal mechanisms (ICM) principle (Schölkopf et al., 2012; Peters et al., 2017) assumes that the data generating process is composed of independent and autonomous modules that do not inform or influence each other. The promising capability of causal mechanisms grows an activate subfield (Parascandolo et al., 2018; Locatello et al., 2018a;b; Bengio et al., 2019). Recent works define the mechanisms to be: 1) functions that generate a variable from the cause (Bengio et al., 2019), 2) functions that transform the data (e.g. rotation) (Parascandolo et al., 2018), and 3) a disentangled mixture of independent generative models that generate data from distinct causes (Locatello et al., 2018a;b). Throughout this paper, we refer to type 2) mechanisms as shared mechanisms and type 3) mechanisms as generative mechanisms.

Despite the recent progress, unsupervised learning of the generative and shared mechanisms from complex observational data (e.g. images) remains a difficult and unsolved task. In particular, previous approaches (Locatello et al., 2018a;b) for disentangling the generative mechanisms rely on competitive training, which does not directly enforce the disentanglement between generative mechanisms. The empirical results show entanglement. Additionally, Parascandolo et al. (2018) proposed a mixture-of-experts-based method to learn the shared mechanisms using a canonical distribution and a reference distribution, which contains the transformed data from the canonical distribution. Such a reference distribution is generally unavailable in real-world datasets. To create a reference distribution, we need to use the shared mechanisms that we aim the learn. This causes a chicken-egg problem. Besides, the unsupervised learning of the deep generative model is proved to be unidentifi-

able (Locatello et al., 2019; Khemakhem et al., 2020). Lacking identifiability makes it impossible to learn the right disentangled model (Locatello et al., 2019). Recent methods (Locatello et al., 2020; Khemakhem et al., 2020) leverage weak-supervision or auxiliary variables to identify the right deep generative model. However, such weak-supervision or auxiliary variables still do not exist in conventional datasets (e.g. MNIST).

We, therefore, seek a practical algorithm with identifiability result that disentangles the mechanisms from i.i.d data without manual supervision. To this end, we propose a single self-supervised generative model with an unconventional mixture prior. In the following sections, we refer to our model as the ICM model. Using a single self-supervised generative model would allow us to leverage the recent progress in deep generative clustering (Mukherjee et al., 2019), which would enforce the disentanglement between the generative mechanisms. We use the following example to illustrate the relationship between the generative model and the mechanisms. Let us assume we have a generative model $G : \mathcal{Z} \rightarrow \mathcal{X}$, two generative mechanisms $M_0 : \mathcal{Z}_{M_0} \rightarrow \mathcal{X}_{M_0}, M_1 : \mathcal{Z}_{M_1} \rightarrow \mathcal{X}_{M_1}$, and one shared mechanism $M_S : \mathcal{X}_M, \mathcal{Z}_S \rightarrow \mathcal{X}$ where $\mathcal{Z} = [\mathcal{Z}_{M_0}, \mathcal{Z}_{M_1}, \mathcal{Z}_S]$ and $\mathcal{X}_M = \mathcal{X}_{M_0} \cup \mathcal{X}_{M_1}$. We have $G([z_{M_0}, \mathbf{0}, z_S]) = M_S(M_0(z_{M_0}), z_S)$ and $G([\mathbf{0}, z_{M_1}, z_S]) = M_S(M_1(z_{M_1}), z_S)$. Our mixture prior is unconventional because the mixture components are $\{[\mathcal{N}(\mathbf{0}, \mathbf{I}), \mathbf{0}, \mathcal{N}(\mathbf{0}, \mathbf{I})], [\mathbf{0}, \mathcal{N}(\mathbf{0}, \mathbf{I}), \mathcal{N}(\mathbf{0}, \mathbf{I})]\}$ instead of $\{\mathcal{N}(\boldsymbol{\mu}_0, \boldsymbol{\sigma}_0^2), \mathcal{N}(\boldsymbol{\mu}_1, \boldsymbol{\sigma}_1^2)\}$. To keep the notations clear, we omit the normalization factor. We disentangle the generative mechanisms by disentangling the type of variations (causes) carried by each $z_{M_k}, \forall k \in \{1, 2, ..., N\}$ where $N$ is the number of generative mechanisms. The disentanglement between the generative mechanisms and the shared mechanisms will be guaranteed by the prior itself. Furthermore, we theoretically prove that the ICM model is identifiable w.r.t. the mechanisms without accessing any label. The key contributions of this paper are:

- We propose a simpler method to learn the mechanisms with only self-supervision.
- We design an unconventional mixture prior that enforce disentanglement.
- We prove the first identifiability result w.r.t. the mechanisms in the self-supervised scenario.
- We develop a novel method to quantitatively evaluate the robustness of ML models under covariant shift using the covariant that is naturally encoded in the data.
- We conduct extensive experiments to show that our ICM model is more robust against intervention, covariant shift, and noise compared to disentangled representations.

## 2 RELATED WORK

**Functional Causal Model**  In functional causal model (FCM), the relationships between variables are expressed through deterministic, functional equations: $x_i = f_i(pa_i, u_i), i = 1, ..., N$. The uncertainty in FCM is introduced via the assumption that variables $u_i, i = 1, ..., N$, are not observed (Pearl et al., 2000). If each function in FCM represents an autonomous mechanism, such FCM is called a structural model. Moreover, if each mechanism determines the value of one and only one variable, then the model is called a structural causal model (SCM). The SCMs form the basis for many statistical methods (Mooij & Heskes, 2013; Mooij et al., 2016) that aim at inferring knowledge of the underlying causal structure from data (Bongers et al., 2016). Taking the view from the SCM's perspective, we want to learn a mixture of causal models whose inputs are pure latent variables and whose output is a single high-dimensional variable that describes complex data such as images. Different from other SCM approaches, where the unobserved variables only introduce uncertainty to the model, the latent variables in our model carries distinct variations in the dataset.

**Independent Component Analysis**  Discovering independent components of the data generating process has been studied intensively (Hyvärinen & Oja, 2000; Hyvarinen et al., 2019). A recent work (Khemakhem et al., 2020) bridges the gap between the nonlinear independent component analysis (ICA) and the deep generative model. The nonlinear ICA with auxiliary variables brings parameter-space identifiability to variational auto-encode. The nonlinear ICA tackles the parameter-space identifiability of deep generative models. However, the parameter-space identifiability does not guarantee the disentanglement between causes. We will discuss the difference in section 4.

**Disentangled Representations**  Disentangled representations assume that the data is generated using a set of independent latent explanatory factors (Bengio et al., 2013). Previous works (Higgins

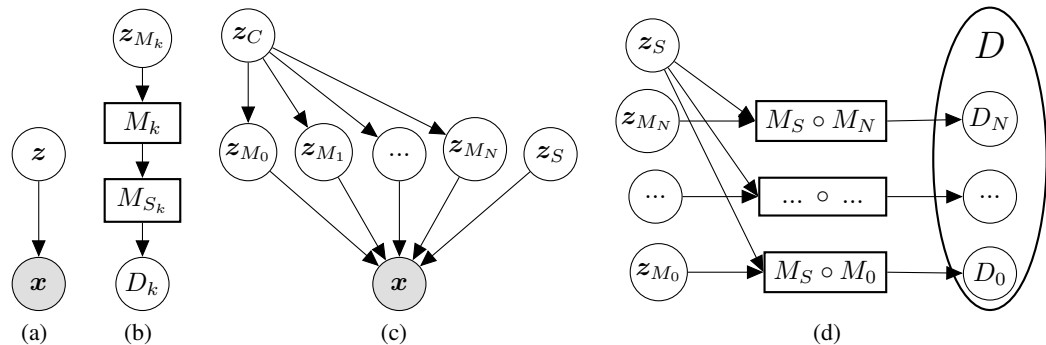

Figure 1: (a) The graphical model of deep generative models and disentangled representations. (b) The assumed data generation process in previous works (Parascandolo et al., 2018; Locatello et al., 2018b), where $z_{M_k}$ is the cause for the $k^{th}$ generative mechanism, $M_k$ is the generative mechanism and $M_{S_k}$ is the shared mechanism. (c) The graphical model of our approach, where $z_C$ is a confounding variable that controls which group of causes will take effect. (d) The assumed data generation processes in our approach. Each cluster $D_k$ in the data set $D$ is generated by its associated generative mechanism $M_k$ and the shared mechanisms $M_S$. $z_{M_k}$ and $z_S$ are the causes for the generative mechanism $M_k$ and the shared mechanism $M_S$, respectively.

et al., 2017; Kumar et al., 2017) proposed various unsupervised methods to disentangle the latent explanatory factors. Later, unsupervised disentanglement is proved to be impossible (Locatello et al., 2019). Recent works (Shu et al., 2019; Locatello et al., 2020) leverage weak-supervision (e.g. pair-wise label) to identify the right disentangled model. Compared to disentangled represtations, our ICM model takes a different assumption. We assume there are groups of latent explanatory factors that generate clusters in the data distribution, respectively.

## 3 GENERATIVE MODEL FOR DISENTANGLED MIXTURE OF MECHANISMS

### 3.1 PRELIMINARIES

Many generative models (Kingma & Welling, 2013; Goodfellow et al., 2014; Locatello et al., 2020) assume that the data $x$ is generated through a two-step procedure: (1) A sample $z$ is drawn from an unobserved continuous prior distribution $p(z)$, which is usually assumed to be $\mathcal{N}(\mathbf{0}, I)$. (2) An observational data sample $x$ is drawn from an unknown conditional distribution $p(x \mid z)$ or is generated by a function $f : \mathcal{Z} \rightarrow \mathcal{X}$. Figure 1(a) and 1(b) further visualize this procedure from a causal mechanisms' perspective.

**Generative Adversarial Network** Generative Adversarial Network (GAN) learns the data generation process via a two-player minmax game between a generator $G : \mathcal{Z} \rightarrow \mathcal{X}$ and a discriminator $D : \mathcal{X} \rightarrow \mathcal{Y}$: $\min_G \max_D V(G, D) = \mathbb{E}_{x \sim p(x)}[\log D(x)] + \mathbb{E}_{z \sim p(z)}[\log(1 - D(G(z)))]$. The goal is to minimize the divergence between the generated data and the real data. Recent research has shown that the Wasserstein distance (Arjovsky et al., 2017) is a good choice of divergence in practice. Thus, we use Wasserstein GAN with gradient penalty (WGAN-GP) (Gulrajani et al., 2017) as our generative model. GAN are less popular in unsupervised learning of disentangled representations as is difficult to approximate the posterior $p(z \mid x)$. However, recent work (Mukherjee et al., 2019) showed its advantages in self-supervised deep clustering, which is highly relevant to disentangling the data generation process. Thus, in addition to the generator network and the discriminator network of a conventional GAN, we add an encoder network to perform self-supervision, which is discussed in section 3.3.

### 3.2 THE UNCONVENTIONAL MIXTURE PRIOR AND STRUCTURAL LATENT SPACE

Unlike most generative models which assume $p(z) = \prod_{i=1}^{d} p(z_i)$, we use an unconventional mixture prior:

$$p(\boldsymbol{z}) = p(\boldsymbol{z}_S) \frac{\sum_{z_C} \prod_{k=1}^{N} p(\boldsymbol{z}_{M_k} \mid z_C)}{Z} = p(\boldsymbol{z}_S) \sum_{k=1}^{N} \frac{p_k(\boldsymbol{z}_M)}{Z}$$

$$p(\boldsymbol{z}_{M_k} \mid z_C) = \left\{ \begin{array}{ll} \mathcal{N}(\mathbf{0}, \boldsymbol{I}) & , z_C = k \\ \mathbf{0} & , \text{otherwise}. \end{array} \right. \tag{1}$$

where $Z = \sum_{k=1}^{N} \int_{\boldsymbol{z}_M} p_k(\boldsymbol{z}_M) d\boldsymbol{z}_M$, $p(\boldsymbol{z}_S) = \prod_{i=1}^{d_S} p(z_{S_i})$, $p(\boldsymbol{z}_{M_k}) = \prod_{i=1}^{d_{M_k}} p(z_{M_{k_i}})$, and $\boldsymbol{z}_M = [\boldsymbol{z}_{M_1}, \boldsymbol{z}_{M_2}, ..., \boldsymbol{z}_{M_N}]$. $\boldsymbol{z}_S$ represents the causes for the shared mechanisms and $\boldsymbol{z}_{M_k}$ represents the causes for the $k^{\text{th}}$ generative mechanisms. We convert the sum of the conditional joint distribution to our unconventional mixture distribution by defining $p_1(\boldsymbol{z}_M) = [\mathcal{N}(\mathbf{0}, \boldsymbol{I}), \mathbf{0}, ..., \mathbf{0}]$, $p_2(\boldsymbol{z}_M) = [\mathbf{0}, \mathcal{N}(\mathbf{0}, \boldsymbol{I}), ..., \mathbf{0}]$ and so on. Figure 1(c) and 1(d) further visualize the graphical model and the assumed data generation process with the unconventional mixture prior.

Our prior disentangles the mixture components by encoding the components in orthogonal latent subspaces. The mixture components of Gaussian mixture prior, however, are shown to entangle in the latent space (Mukherjee et al., 2019) because it is perfectly fine for mixture components $\mathcal{N}(\boldsymbol{\mu}_0, \boldsymbol{\sigma}_0^2)$ and $\mathcal{N}(\boldsymbol{\mu}_1, \boldsymbol{\sigma}_1^2)$ to have overlap. Such an overlap entangles the causes of different mechanisms. For the shared mechanisms, we encode all causes, such as rotation and brightness, in $\boldsymbol{z}_S$ and do not further disentangle them because such disentanglement is proved to be impossible without supervision (Locatello et al., 2019). Although we do not disentangle each shared cause, it usually does not hurt the performance of downstream tasks because most of the predictive tasks are associated with the generative mechanisms. Thus, we only need to disentangle the shared mechanisms from the generative mechanisms. In section 4, we show such disentanglement is guaranteed, provably.

Besides the Gaussian mixture prior, we also discuss the advantage of our unconventional mixture prior over the existing disentangled prior (Higgins et al., 2017; Locatello et al., 2020). To see this, consider that by definition, each $z_i$ in disentangled prior $\boldsymbol{z}$, which admits a density $p(\boldsymbol{z}) = \prod_{i=1}^{d} p(z_i)$, represents one type of variations in disentangled representations. With the disentangled prior, we can generate data through the generative model by setting each $z_i$ to an arbitrary value. If we use each $z_i$ to represent each cause, the prior would allow the generative mechanism to generate data by arbitrarily combining the causes. This contradicts our assumption that each cluster in the data distribution has its distinct cause and none of the data is generated by the combined causes. To resolve this conflict, we need to either create a new dataset or adopt the unconventional mixture prior.

### 3.3 THE SELF-SUPERVISED MODEL

Our model is composed of a generator network $G : \mathcal{Z} \to \mathcal{X}$, a discriminator network $D : \mathcal{X} \to \mathcal{Y}$, and an auxiliary encoder network $E : \mathcal{X} \to \mathcal{Z}$. The auxiliary encoder is a basic deterministic encoder that tries to invert the generator. The self-supervision means predicting which generative mechanism does the generated data come from using the auxiliary encoder network $E$. Such self-supervision would encourage the data samples from the same generative mechanism to be similar to each other and vice versa. Thus, we can disentangle generated data from different mechanisms in the sense that they are separable and their sources are predictable. Once the data samples are disentangled into clusters, the generative mechanisms are disentangled as well. The loss function of our method is:

$$\mathcal{L} = W(\mathbb{P}_g || \mathbb{P}_r) + \beta_C \mathbb{E}_{\boldsymbol{z} \sim P(\boldsymbol{z})} \mathcal{H}(\boldsymbol{z}_C, E(G(\boldsymbol{z}))_C) + \beta_r \mathbb{E}_{\boldsymbol{x} \sim p(\boldsymbol{x})} ||\boldsymbol{x} - G(E(\boldsymbol{x}))||_2^2$$
$$+ \beta_M \mathbb{E}_{\boldsymbol{z} \sim p(\boldsymbol{z})} ||\boldsymbol{z}_M - E(G(\boldsymbol{z}))_M||_2^2 + \beta_S \mathbb{E}_{\boldsymbol{z} \sim p(\boldsymbol{z})} ||\boldsymbol{z}_S - E(G(\boldsymbol{z}))_S||_2^2 \tag{2}$$

where $\mathbb{P}_g$ denotes the distribution covered by the generator network $G$, $\mathbb{P}_r$ denotes the real data distribution, $W(\mathbb{P}_g || \mathbb{P}_r)$ denotes the Wasserstein distance between two distributions, $\boldsymbol{z} = [\boldsymbol{z}_C, \boldsymbol{z}_M, \boldsymbol{z}_S]$, $\boldsymbol{z}_C$ is a categorical variable represents the index of the generative mechanisms, $\boldsymbol{z}_M$ and $\boldsymbol{z}_S$ are the same as is defined in section 3.2, and $\mathcal{H}$ denotes the cross-entropy loss. We have tried to replace the

Euclidean distance with Cosine distance in the loss function, but both methods yield similar results. As the readers may have noticed, we add $z_C$ to $z$ in our implementation. The purpose is to eliminate the ambiguity when $z_M \approx 0$ and the generator does not know which generative mechanism to use (Antoran & Miguel, 2019). As we are only fixing a corner case, we leave it as an implementation detail and keep the unconventional mixture prior definition unchanged.

The loss function above can be decomposed into four parts: 1) $W(\mathbb{P}_g || \mathbb{P}_r)$ is the WGAN loss. 2) $\mathbb{E}_{z \sim P(z)} \mathcal{H}(z_C, E(G(z))_C)$ is the self-supervision loss that enforces the disentanglement between generated data. 3) $\mathbb{E}_{x \sim p(x)} ||x - G(E(x))||_2^2$ is the forward cycle-consistent loss, which enforces $x \approx G(E(x))$. The encoder takes the generated data from the generator as input. However, the generated data distribution may differ from the real data distribution. Thus, certain data in the real distribution may become out-of-distribution (OOD) for the encoder. Such distribution divergence hurts the performance of downstream tasks. The forward cycle-consistency mitigates the distribution divergence by encouraging the ICM model to cover the whole real distribution. 4) $\mathbb{E}_{z \sim p(z)} ||z_M - E(G(z))_M||_2^2 + \beta_S \mathbb{E}_{z \sim p(z)} ||z_S - E(G(z))_S||_2^2$ is the backward cycle-consistent loss, which enforces $z \approx E(G(z))$. The backward cycle-consistency prevents the generator from discarding certain causes in the generation process.

## 4 IDENTIFIABLE DISENTANGLEMENT

Identifiability is crucial for disentangling the causal mechanisms. We can construct infinitely many generative models that have the same marginal distribution (Locatello et al., 2019; Khemakhem et al., 2020). Without identifiability, any one of these models could be the true causal generative model for the data, and the right model cannot be identified given only the observational data (Peters et al., 2017). Prior work (Locatello et al., 2019) shows that we can not identify the disentangled generative model using the marginal distribution. On learning the causal mechanisms, we are interested in identifying the right generative model that disentangles the causes in the latent space. Specifically, each $z_{M_k}$ and $z_S$ should represent distinct causes when $z$ is paired with the right generative model $g$. Before beginning the proof, we introduce the impossible result in unsupervised learning of disentangled representations, where they aim to let each latent explanatory factor $z_i \in z$ carries a distinct type of variations.

**Theorem 1.** *(Locatello et al., 2019) For $d > 1$, let $z \sim P$ denote any distribution which admits a density $p(z) = \prod_{i=1}^{d} p(z_i)$. Then, there exists an infinite family of bijective functions $h : \text{supp}(z) \to \text{supp}(z)$ such that $\frac{\partial h_i(u)}{\partial u_j} \neq 0$ almost everywhere for all $i$ and $j$ (i.e., $z$ and $h(z)$ are completely entangled) and $P(z \leq u) = P(h(z) \leq u)$ for all $u \in \text{supp}(z)$ (i.e., they have the same marginal distribution).*

Theorem 1 indicates that we can not identify the disentangled model because we can not tell whether the latent explanatory factors are entangled by $h$ or not from the marginal distribution. Compared to the parameter-space identifiability in nonlinear ICA (Khemakhem et al., 2020), identifying a disentangled model is more challenging because even if we can identify the right distribution, the function $h$ may still entangle the latent explanatory factors. Another difference is that the ground-truth distribution of the latent variables in disentangled representations is assumed to be isotropic Gaussian, which diminishes the importance of the parameter-space identifiability. Therefore, we formulate our identifiability in the function space:

**Definition 1.** *Let $\sim_h$ be an equivalence relation on $g : \mathcal{Z} \to \mathcal{X}$ as follows:*

$$g \sim_h \tilde{g} \Leftrightarrow g^{-1}(x) = h(\tilde{g}^{-1}(\tilde{x})), \forall x \in \mathcal{X} \tag{3}$$

*where $h : \mathcal{Z} \to \mathcal{Z}$ is a smooth invertable function.*

**Definition 2.** *We say that $g : \mathcal{Z} \to \mathcal{X}$ is identifiable up to $\sim_h$ (or $\sim_h$-identifiable) if:*

$$p_g(x) = p_{\tilde{g}}(x) \Rightarrow g \sim_h \tilde{g} \tag{4}$$

Under the definitions 1 and 2, the following lemma shows that the generative model $g$ is $\sim_h$-identifiable:

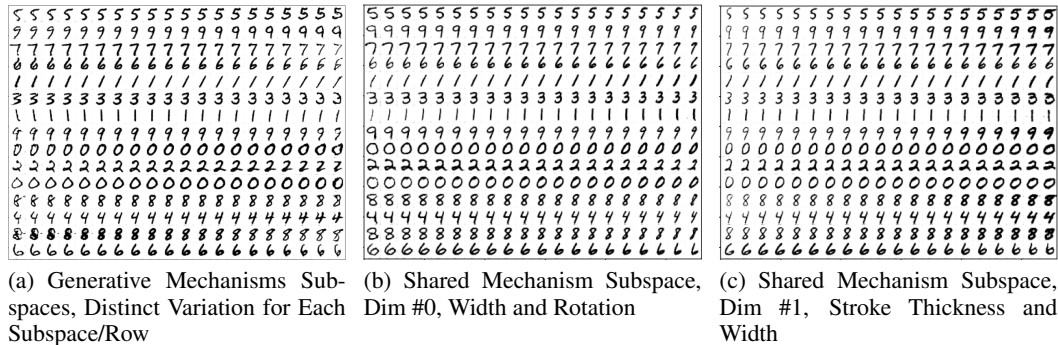

(a) Generative Mechanisms Sub-spaces, Distinct Variation for Each Subspace/Row

(b) Shared Mechanism Subspace, Dim #0, Width and Rotation

(c) Shared Mechanism Subspace, Dim #1, Stroke Thickness and Width

Figure 2: Latent Space Traversal of ICM model on MNIST

**Lemma 2.** *Let $\mathcal{G}$ be the space of smooth invertable functions with smooth inverse (i.e., a diffeomorphism) that map $\mathcal{Z}$ to $\mathcal{X}$, and $h : \mathcal{Z} \to \mathcal{Z}$ is a smooth invertable function. Then, any function $g \in \mathcal{G}$ can be represented as $g = g^* \circ h$, where $g^* : \mathcal{Z} \to \mathcal{X}$ is the assumed ground-truth disentangled generative model in the function space $\mathcal{G}$. Formally, we have $g \sim_h g^*, \forall g \in \mathcal{G}$ and the model $g$ is $\sim_h$-identifiable.*

The proof is in Appendix A. The following theorem further shows that the function $h$ in $\sim_h$ disentangles the groups of causes for different mechanisms with the unconventional mixture prior.

**Theorem 3.** *Let $p(\boldsymbol{z})$ denote the unconventional mixture prior. We use $\mathcal{M}$ to denote the manifold where the ground-truth latent variable $\boldsymbol{z}$ lies on. Let $k \in \{1, 2, ..., N\}$ be the index of generative mechanisms $M$. We assume each $\boldsymbol{z}_{M_k}$ lies on $\mathcal{M}_{M_k}$ and $\boldsymbol{z}_S$ lies on $\mathcal{M}_S$. We let $\hat{\boldsymbol{z}} = h(\boldsymbol{z})$ which lies on $\hat{\mathcal{M}}$. $g^*$ is the ground-truth disentangled model. Then, if there exists an smooth invertable function $h : \mathcal{Z} \to \mathcal{Z}$ such that $g = g^* \circ h$ maps $\mathcal{Z}$ to $\mathcal{X}$, we have $h$ maps each $\mathcal{M}_{M_k}$ to disjoint sub-manifold $\hat{\mathcal{M}}_{M_k}$ and maps $\mathcal{M}_S$ to $\hat{\mathcal{M}}_S$, which is disjoint from all $\hat{\mathcal{M}}_{M_k}$.*

We leave the proof of Theorem 3 in Appendix B. By a slight abuse of notation, we define the disentangled $h$ as $h_D$. Then, the disentangled ICM model is $\sim_{h_D}$-identifiable.

## 5 EXPERIMENT

For various downstream tasks, we use the encoder network $E$ to extract data representations. In this section, we first examine the robustness of our model against interventions. As we only apply interventions on the root node, such interventions are equivalent to conditional generation. We visualize the result by latent space traversal. We then evaluate the robustness of our model under the covariant shift by measuring the downstream ML model performance. The covariate shift is the change in the distribution of the covariates, that is, the independent variables. More specifically, we consider disentangled single covariant shifts (e.g. rotation) and entangled multiple covariant shifts (e.g. brightness and width). Unlike previous works (Arjovsky et al., 2019; Locatello et al., 2020), which rely on manual annotation, we propose a new method to extract the latent covariants from the datasets and use them to create experiments. Finally, we measure the robustness of our model against uniform noise and Gaussian noise by measuring the downstream ML model performance. Through our experiments, we use: 1) Two datasets, which are MNIST and FashionMNIST. 2) Three competitors, which are VAE (Kingma & Welling, 2013), $\beta$-VAE (Higgins et al., 2017), and Ada-GVAE (Locatello et al., 2020). 3) Four covariant shifts, which are implicitly carried by the datasets.

### 5.1 INTERVENTIONAL ROBUSTNESS AND LATENT SPACE TRAVERSAL

We show the latent space traversal of the causal mechanisms on MNIST in Figure 2. In this experiment, each generative mechanism $M_k$ has an one dimensional $z_{M_k}$. The shared mechanisms has a four dimensional $\boldsymbol{z}_S$. For the $k^{\text{th}}$ generative mechanism subspace traversal, we set the $\boldsymbol{z}_C$ to $k$ and manually change $z_{M_k}$. During this process, all the irrelevant dimensions are set to 0. As Figure

Table 1: Average Accuracy of Downstream Classifiers under Different Shift Strength

| MODEL | MNIST (T) | MNIST (W&R) | MNIST (R) | FASHIONMNIST (D&W) |
|-------|-----------|-------------|-----------|---------------------|
| ICM | **74.63%** | **56.91%** | 43.71% | **42.92%** |
| VAE | 55.91% | 41.92% | 39.90% | 37.08% |
| $\beta$-VAE | 73.34% | 46.33% | 46.05% | 41.60% |
| ADA-GVAE | 68.09% | 47.16% | **47.58%** | 40.85% |

Table 2: Normalized Accuracy Variations of Downstream Classifiers under Different Shift Strength

| MODEL | MNIST (T) | MNIST (W&R) | MNIST (R) | FASHIONMNIST (D&W) |
|-------|-----------|-------------|-----------|---------------------|
| ICM | **2.43%** | **3.70%** | **7.10%** | **5.02%** |
| VAE | 4.32% | 6.30% | 7.33% | 9.02% |
| $\beta$-VAE | 2.71% | 4.98% | 7.80% | 7.83% |
| ADA-GVAE | 2.96% | 4.90% | 7.31% | 7.98% |

Table 3: Shift Distance Needed for 10% Relative Accuracy Drop under Covariant Shift

| MODEL | MNIST (T) | MNIST (W&R) | MNIST (R) | FASHIONMNIST (D&W) |
|-------|-----------|-------------|-----------|---------------------|
| ICM | **1.4** | **0.8** | **0.6** | **0.4** |
| VAE | 0.6 | 0.4 | 0.4 | 0.2 |
| $\beta$-VAE | 1.1 | 0.5 | 0.4 | 0.3 |
| ADA-GVAE | 1.1 | 0.5 | 0.4 | 0.3 |

2(a) shows, each row represents a traversal of an generative mechanism subspace. For the shared mechanism, we first let $z_C = 0$. Then, we do a traversal for a dimension in the shared mechanism subspace. After this traversal, we increase $z_C$ by 1 and repeat the procedure. In Figures 2(b) and 2(c), each figure represents a traversal for a dimension in the shared mechanism subspace and each row in the same figure represents a traversal of a fixed dimension with different $z_C$. Due to the limited space, we report more visualization results in the Appendix.

We say our ICM model is robust against intervention in the sense that no matter how we change $z_{M_k}$ and $z_S$ while keeping $z_C$ and $z_{\setminus M_k}$ fixed, the generator does not generate data that does not belong to mechanism $M_k$ (e.g. digit 0 does not change to other digits during the traversal). Furthermore, we show that the traversal of each generative mechanism yields a distinct, mechanisms-specific type of variation. The traversal of the shared mechanism yields the same type of variation. Compared with conditional generative models (Kingma et al., 2014; Klys et al., 2018), our work do not pose any requirement on the label during training.

Finally, we note that the number of generative mechanisms is chosen to be larger than the number of classes, which means one ground-truth mechanism may be split into two estimated mechanisms. Previous works (Parascandolo et al., 2018; Locatello et al., 2018b) also adopt the same setting and we do not find it triggers any issue in the following experiment.

### 5.2 ROBUSTNESS UNDER COVARIANT SHIFT

To quantitatively measure the robustness under the covariant shift, we first partition the dataset using the learned representations as the covariant. For convince, we use the encoder of the NVAE model (Antoran & Miguel, 2019) to extract the representations. Using the shared mechanism in our ICM model would yield the same result. More specifically, we partition the dataset into subsets $\{x \mid x \in \mathcal{X}, \hat{z}_i = E(x)_i \in [C_{lower}, C_{upper})\}$ using the predicted value in dimension $i$. For the training set, we set $[C_{lower}, C_{upper})$ to $[0, \infty)$. For each test set, we set $[C_{lower}, C_{upper})$ to $[-0.1, 0), [-0.2, -0.1), ..., [-3.0, -2.9)$. We use the $C_{lower}^{train} - C_{upper}^{test}$ to represent the strength of the covariant shift, which is also called the shift distance in Table 3. In this experiment, we consider

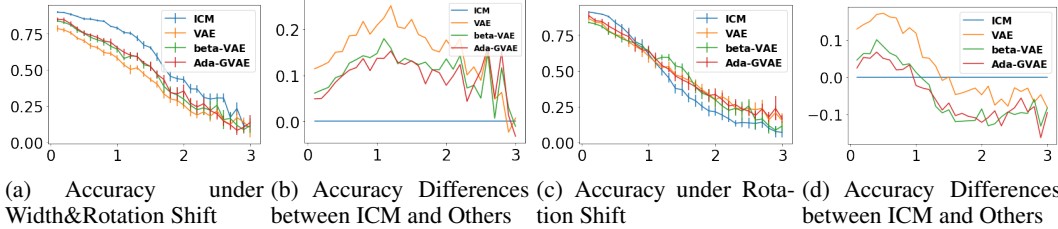

(a) Accuracy under Width&Rotation Shift  (b) Accuracy Differences between ICM and Others  (c) Accuracy under Rotation Shift  (d) Accuracy Differences between ICM and Others

Figure 3: Accuracy and Accuracy Differences of Downstream Classifiers under covariant Shift on MNIST dataset. If ICM outperforms another method, the accuracy difference is positive.

Table 4: Accuracy of Downstream Classifiers under Noise $\mathcal{N}(\mathbf{0}, \boldsymbol{I})$

| MODEL | MNIST (T) | MNIST (W&R) | MNIST (R) | FASHIONMNIST (D&W) |
|---|---|---|---|---|
| ICM | **52.64%** $\pm$ **2.03%** | 39.36% $\pm$ 2.58% | **53.30%** $\pm$ **2.08%** | **46.69%** $\pm$ **1.75%** |
| VAE | 34.13% $\pm$ 1.48% | 32.40% $\pm$ 2.07% | 27.14% $\pm$ 2.32% | 33.19% $\pm$ 2.52% |
| $\beta$-VAE | 39.98% $\pm$ 2.74% | 39.93% $\pm$ 2.23% | 37.35% $\pm$ 3.16% | 34.79% $\pm$ 2.86% |
| ADA-GVAE | 46.41% $\pm$ 1.66% | **43.10%** $\pm$ **0.86%** | 46.25% $\pm$ 0.77% | 43.92% $\pm$ 1.61% |

stroke thickness (T), width and rotation (W&R), rotation (R) and, darkness and width (D&W) as covariants. Figure 4 in the Appendix further visualizes these covariants.

Then, we use a gradient boosting classifier (GBT) from Scikit-learn with default parameters as the downstream ML model, which is the same as previous works (Locatello et al., 2019; 2020). We train the classifier with 1000 samples, which is sufficient for producing good accuracy, and test its accuracy on a sequence of test sets. For each experiment, we evaluate the classifier 10 times and collect the average accuracy as well as the accuracy variations. Tables 1 and 2 show the average of average accuracy and the average of normalized variation of the classifier using each model's representation as input under different covariant shift strengths, respectively. Our ICM model achieves the best average accuracy across all the experiments except for MNIST (R). For the normalized accuracy variations, which is the standard deviation over the accuracy, our ICM model always achieves the lowest variation. It means that the ICM model is less sensitive to the choice of training samples as the GBT classifier does not contribute many variations.

We further investigate the issue of our model under MNIST (R) covariant shift. Figure 3 shows the accuracy changes as the shift strength increases. In both experiments, our method shows more robustness when the shift strength is low as its accuracy decreases slower and its advantage grows bigger. However, after a threshold, our method begins to lose its advantage. There are two possible reasons: 1) The test data shifts too far from the training data and the base generative model can not generalize to the test sets. After the test set shifts too far away, none of the methods perform well. It's hard to conclude that a model with $\sim 40\%$ accuracy is better than a model with $\sim 35\%$ accuracy. 2) The test set there contains too few samples, which is just around tens or hundreds, and makes the evaluation inaccurate. As we can see from Figure 3, the larger the shift, the bigger the accuracy variations. To eliminate this interference, we instead measure how much distance the covariant shift needs to go to decrease the accuracy by a percentage relatively. Such evaluation will put more weight on the test sets which have more samples and yield reasonable accuracy. Table 3 and Tables 9, 10 in Appendix D.2 show our method can tolerate more covariant shift before the accuracy relatively drops by 10%, 20%, and 40%.

## 5.3 ROBUSTNESS AGAINST NOISE

We evaluate the robustness of our ICM model under noise using a similar setting as the previous section. In this experiment, we use the train the GBT with 1000 samples and test the GBT accuracy on whole train set. Table 5.3 shows our ICM model is generally more robust against Gaussian noise

Table 5: Accuracy of Downstream Classifiers under Noise $\epsilon * \mathcal{N}(\mathbf{0}, \boldsymbol{I})$ on MNIST (W&R)

| MODEL | $\epsilon = 0.25$ | $\epsilon = 0.5$ | $\epsilon = 0.75$ |
|---|---|---|---|
| ICM | $\mathbf{85.04\% \pm 0.57\%}$ | $\mathbf{69.59\% \pm 2.05\%}$ | $51.61\% \pm 4.68\%$ |
| VAE | $74.97\% \pm 0.66\%$ | $58.57\% \pm 1.44\%$ | $42.90\% \pm 1.42\%$ |
| $\beta$-VAE | $80.08\% \pm 0.77\%$ | $64.94\% \pm 2.53\%$ | $49.51\% \pm 2.02\%$ |
| ADA-GVAE | $81.64\% \pm 1.29\%$ | $68.38\% \pm 1.76\%$ | $\mathbf{53.90\% \pm 1.68\%}$ |

$\mathcal{N}(\mathbf{0}, \boldsymbol{I})$. Table 5.3 further shows that although there are methods that perform slightly better than our method on MNIST (W&R), our method still performs better before the noise becomes too large and decrease the accuracy by a high percentage. We further show the results under uniform noise Uniform$(\mathbf{0}, \boldsymbol{I})$ in Appendix D.3.

## 6 CONCLUSION

In this paper, we present a self-supervised method to learn the independent causal mechanisms. We show the presents of generative mechanisms and shared mechanisms. Our model can learn these two types of mechanisms through an unconventional mixture prior. Furthermore, we outline the sufficient conditions for theoretically identifying the mechanisms from observational data. Experiments show that our ICM model is generally more robust against interventions, covariant shifts, and noise.

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
