# OpenReview forum: "Identifying Coarse-grained Independent Causal Mechanisms with Self-supervision"
_ICLR.cc/2021/Conference — Reject_

### Official Review · AnonReviewer1 · 2020-10-28
**a new method for disentangling independent causal mechanisms**

**Rating:** 5
**Confidence:** 4

**Review:**

Authors propose a new method to learn independent causal mechanisms from data, which is achieved by designing a mixture prior consisting of shared mechanisms and Independent Causal Mechanisms (ICM) conditioned mechanisms. Specific orthogonal structure or separability is imposed on the ICM-conditioned mechanisms. A single generative model is used with such a mixture prior to learn mechanisms from data. Authors also prove the subspace of the mechanisms are separated and thus identifiable. Experiments on MNIST dataset demonstrate that the method is able to learn independent mechanisms and it improves the robustness against intervention, co-variant shift and noise.

Detailed Comments:
1. I am not quite clear about the meaning of ICM-conditioned mechanisms. Could authors elaborate it a little bit what “conditioned” means.

2. The Independent Causal Mechanisms first proposed by (Scholkopf, 2019) is referring to the independence of each sub-modules in a whole generating process, e.g. for a cause-effect pair (x, y), ICM usually means that the process of generating the cause which is p(x) is independent of the process that maps the cause to the effect which is p(y|x). However, in this paper, authors are considering a very different setting where they argue that for different data groups, the generative mechanisms are different. I am not very sure they are the same thing. Could authors explain the connection between the two?

3. Authors use single generative model which is claimed to be one of the advantage of their method. However, in Fig 1(d) they use M_N, …, M_0 to indicate that for different data groups, the generating mechanisms are different. Could authors explain this discrepancy?

4. It appears to me that the method can only work in the scenario where we know beforehand the number of different generating mechanisms in the data. However, for real applications it is difficult to have such prior information. Can authors also comment about the applicability of the method?

5. It appears to me that the proposed method is incremental to existing disentangled representation learning methods except for the mixture prior.

---

> ### Author Response · Authors · 2020-11-21
> **[Rebuttal 2/2] Assumption justified; Method specified; Contribution justified**
>
> **Reference:**
>
> [1] Schölkopf, Bernhard, et al. "On causal and anticausal learning." arXiv preprint arXiv:1206.6471 (2012).
>
> [2] Peters, Jonas, Dominik Janzing, and Bernhard Schölkopf. Elements of causal inference. The MIT Press, 2017.
>
> [3] Parascandolo, Giambattista, et al. "Learning independent causal mechanisms." International Conference on Machine Learning. PMLR, 2018.
>
> [4] Locatello, Francesco, et al. "Clustering meets implicit generative models." (2018).
>
> [5] Locatello, Francesco, et al. "Competitive training of mixtures of independent deep generative models." arXiv preprint arXiv:1804.11130 (2018).
>
> [6] Locatello, Francesco, et al. "Challenging common assumptions in the unsupervised learning of disentangled representations." international conference on machine learning. PMLR, 2019.
>
> [7] Khemakhem, Ilyes, et al. "Variational autoencoders and nonlinear ica: A unifying framework." International Conference on Artificial Intelligence and Statistics. 2020.

---

> ### Author Response · Authors · 2020-11-21
> **[Rebuttal 1/2] Assumption justified; Method specified; Contribution justified**
>
>
> We thank the reviewer for the constructive feedback on our paper!
>
> **Response to detailed comments:**
>
> $\bullet\ $ I am not quite clear about the meaning of ICM-conditioned mechanisms
>
> $-$ The ICM-conditioned mechanisms refer to the mechanisms that generate a specific data cluster. In contrast, the shared mechanisms apply to all the clusters. We have renamed the “ICM-conditioned mechanism” as “generative mechanism” in the revised paper to avoid confusion.
>
>
> $\bullet\ $ ICM usually means that the process of generating the cause which is p(x) is independent of the process that maps the cause to the effect which is p(y|x). However, authors are considering a very different setting where they argue that for different data groups, the generative mechanisms are different.
>
> $-$ Our setting is consistent with the definition and prior works. The ICM is defined as autonomous modules in the data generation process [1, 2]. It is not restricted to be a specific function class. One example of such a module is a function $f: \mathbf{z}  \xrightarrow{} \mathbf{x}$, where $\mathbf{z}$ is the cause. There are also other example such as image rotation mechanism $f: \mathbf{x}  \xrightarrow{} \mathbf{x}^T$ [3]. Additionally, due to the increasing complexity of the data, a single variable (e.g. image) may follow a mixture distribution where each cluster has distinct causes. For example, the generative mechanism for digit 7 includes a cause to control whether the 7 has a horizontal line crossing its stem. The generative mechanism for digit 9 has a cause to control the circle size. As these two digits are associated with different causes, it is natural to assume they are generated by different mechanisms. Prior works [3, 4, 5] (from the same author as the reviewer's reference) on learning ICM from image data also adopt the same assumption.
>
> $\bullet\ $ The relationship between the generative model and mechanisms are unclear.
>
> $-$ We have one generative model that encodes all the mechanisms. We show the relationship between the generative model and mechanisms as follows:
>
> Let us assume we have a generative model $G: \mathcal{Z} \xrightarrow[]{} \mathcal{X}$, two generative mechanisms $M_0: \mathcal{Z}_{M_0} \xrightarrow{} \mathcal{X}_{M_0}$, $M_1: \mathcal{Z} _{M_1}\xrightarrow{} \mathcal{X} _{M_1}$ and one shared mechanism $M_S: \mathcal{X} _{M_\ }, \mathcal{Z}_S \xrightarrow{} \mathcal{X}$ where $\mathcal{Z} = [\mathcal{Z} _{M_0},\mathcal{Z} _{M_1}, \mathcal{Z}_S]$ and $\mathcal{X} _{M_\ } \= \mathcal{X} _{M_0} \cup \mathcal{X} _{M_1}$. We have $G([\mathbf{z} _{M_0}, \mathbf{0}, \mathbf{z} _{S_\ }]) = M_S(M_0(\mathbf{z} _{M_0}), \mathbf{z} _{S_\ })$ and $G([\mathbf{0}, \mathbf{z} _{M_1}, \mathbf{z} _{S_\ }]) = M_S(M_1(\mathbf{z} _{M_1}), \mathbf{z} _{S_\ })$. We have added this example to the third paragraph of the revised paper.
>
> $\bullet\ $ It appears to me that the method can only work in the scenario where we know beforehand the number of different generating mechanisms in the data
>
> $-$ Our method does not require the users to know the precise number of mechanisms in advance. In terms of applicability, our method has the same applicability as conventional clustering algorithms and the previous ICM method [3]. Our ICM model can tolerate the discrepancy between the learned mechanisms and the ground-truth mechanisms if the number of learned mechanisms is greater or equal to the number of ground-truth mechanisms. This is consistent with our experiment where we use 15 mechanisms on MNIST. The users can easily decide the number of mechanisms by applying the following procedure: 1) Pick a random number and train the ICM model. 2) Do a latent space traversal and observe the type of variations. 3) If the data samples from the same generative mechanism share the same type of variations, the ICM model is ready to use. Otherwise, increase the number of mechanisms and repeat steps 1)&2).
>
> $\bullet\ $ the proposed method is incremental to existing disentangled representation learning methods except for the mixture prior.
>
> $-$ Our proposed method is not incremental because we don't only contribute the mixture prior, but our novel contribution is the identifiability as is discussed in section 4. Without identifiability, it is impossible to achieve disentanglement [6]. To our best knowledge, our coarse-grained identifiability result is the first result that requires no manual supervision. Previously, unsupervised disentanglement with the isotropic Gaussian prior is proved to be unidentifiable [6]. The identifiability result for a mixture of the exponential family requires auxiliary variables [7].

---

### Official Review · AnonReviewer3 · 2020-10-28
**This work assumes the disentanglement of the generation process, with each process corresponding to each cluster of data. The whole methodology is standing on this assumption, which is unreasonable for me.**

**Rating:** 2
**Confidence:** 4

**Review:**

I hardly agree with the assumption made in the graphical model c) and d) that each cluster of data has a different generating mechanism from others. First, it is named the ICM-conditional mechanism in the paper, which is very confusing since it contradicts the definition in the literature of ICM [1]. The ICM, in the literature, originally describes the independent autonomous for generating a sequence of variables in the causal graph, e.g., p(v_1,...,v_k) = \Pi_k p(v_k | Pa(k)), based on the assumption that the exogenous variables are independent with each other. Besides, the difference in terms of variation for digits 1 and 2, in the example made for supporting this assumption, should be encoded in the latent variables z, rather than the generating process. Since the latent variable z is defined to describe the high-level abstractions or concepts, including but not limited to the thickness, width, length in the example of MNIST. Therefore, the graphical model should be y -> z -> x, which is aligned with existing literature in nonlinear ICA.  Besides, even if the assumption is correct, the posterior distribution of p(z_Mk | x_Mk) should be varied across k, which hence cannot be learned by the same encoder E.

Although the experimental results yield good results, it can not be interpreted in this way. The high-level spirit of this paper is based on an invalid assumption, which makes the promising results in experimental parts non-important.


[1] Schölkopf, Bernhard. "Causality for machine learning." arXiv preprint arXiv:1911.10500 (2019).

---

> ### Author Response · Authors · 2020-11-21
> **Assumption justified**
>
> We thank the reviewer for the feedback.
>
> **Response:**
>
> $\bullet\ $ The assumption of data generation processes is unreasonable.
>
> $-$ Our setting is consistent with the definition and prior works. The ICM is defined as autonomous modules in the data generation process [1, 2]. It is not restricted to a specific function class. One example of such a module is a function $f: \mathbf{z}  \xrightarrow{} \mathbf{x}$, where $\mathbf{z}$ is the cause. There are also other examples such as image rotation mechanism $f: \mathbf{x}  \xrightarrow{} \mathbf{x}^T$ [3]. Additionally, due to the increasing complexity of the data, a single variable (e.g., image) may follow a mixture distribution where each cluster has distinct causes. For example, the generative mechanism for digit 7 includes a cause to control whether the 7 has a horizontal line crossing its stem. The generative mechanism for digit 9 has a cause to control the circle size. As these two digits are associated with different causes, it is natural to assume they are generated by different mechanisms. Prior works( [3, 4, 5], from the same author as the reviewer's reference) on learning ICM from image data also adopt the same assumption.
>
> $\bullet\ $ the difference in terms of variation for digits 1 and 2, in the example made for supporting this assumption, should be encoded in the latent variables z
>
> $-$ We did encode the variations in z. The reviewer may have misinterpreted our example. The example states that digits 1 and 2 have different types of variations. The variations are associated with different causes. The goal of disentangled representation learning is to encode each type of variation in each dimension of the latent space.  Therefore, we were saying that if there are two distinct types of variations for digits 1 and 2, the variations should be encoded in different dimensions of z. More specifically, we should at least have a two-dimensional z = [z_0, z_1] for digits 1 and 2 instead of a one-dimensional z. Then, we assign two disentangled data generation processes M_1 and M_2 for z_1 and z_2.
>
> $\bullet\ $ the graphical model should be y -> z -> x
>
> $-$ Our graphical model is y (z_c) -> z -> x with an unconventional mixture prior. We rewrite our prior as follows:
>
> $p(\mathbf{z}) = p(\mathbf{z}_{S}) \frac{\sum _{z_C} \prod _{k=1}^{N} p(\mathbf{z} _{M_k} \mid z_C)}{Z} = p(\mathbf{z}_S) \sum _{k=1}^{N} \frac{p_k(\mathbf{z}_M)}{Z}$
>
> $p(\mathbf{z} _{M_k} \mid z_C) = \mathcal{N}(\mathbf{0}, \mathbf{I}), z_C = k$
>
> $p(\mathbf{z} _{M_k} \mid z_C) = \mathbf{0} , \mathrm{otherwise.}$
>
> where $Z = \sum_{k=1}^{N} \int_{\mathbf{z}_M} p_k(\mathbf{z}_M) d\mathbf{z}_M$. We convert the sum of the joint conditional distribution to our unconventional mixture distribution by defining $p_1(\mathbf{z}_M) = [\mathcal{N}(\mathbf{0}, \mathbf{I}), \mathbf{0}, ..., \mathbf{0}]$, $p_2(\mathbf{z}_M) = [\mathbf{0}, \mathcal{N}(\mathbf{0}, \mathbf{I}), ..., \mathbf{0}]$ and so on.
>
> $\bullet\ $ which is aligned with existing literature in nonlinear ICA
>
> $-$ Our work is not directly comparable to non-linear ICA because the nonlinear ICA can not identify the disentangled model. The nonlinear ICA can identify the latent variable by up to a linear transformation and a shift. The linear transformation will make it impossible to identify the disentangled model because the linear transformation can arbitrarily entangle the latent explanatory factors. This is exactly what we have shown in Theorem 1. Besides, we assume there is no supervision or auxiliary variable which is required by nonlinear ICA.
>
> $\bullet\ $ even if the assumption is correct, the posterior distribution of p(z_Mk | x_Mk) should be varied across k, which hence cannot be learned by the same encoder E.
>
> $-$ Note that our encoder is not a variational auto-encoder (VAE) that computes a posterior, and we have never mentioned "variational" in the paper. The generative model in our paper is a GAN which directly maps the prior to the data distribution. Generally, for a VAE model, it is natural for the encoder to encode each x_Mk cluster at different locations in the latent space. See figure 4(b) in [6] for an example.
>
> **Reference:**
>
> [1] Schölkopf, Bernhard, et al. "On causal and anticausal learning." arXiv preprint arXiv:1206.6471 (2012).
>
> [2] Peters, Jonas, Dominik Janzing, and Bernhard Schölkopf. Elements of causal inference. The MIT Press, 2017.
>
> [3] Parascandolo, Giambattista, et al. "Learning independent causal mechanisms." International Conference on Machine Learning. PMLR, 2018.
>
> [4] Locatello, Francesco, et al. "Clustering meets implicit generative models." (2018).
>
> [5] Locatello, Francesco, et al. "Competitive training of mixtures of independent deep generative models." arXiv preprint arXiv:1804.11130 (2018).
>
> [6] Kingma, Diederik P., and Max Welling. "Auto-encoding variational bayes." arXiv preprint arXiv:1312.6114 (2013).

---

### Official Review · AnonReviewer4 · 2020-10-29
**Interesting paper, but clarity and exposition need to be improved**

**Rating:** 5
**Confidence:** 4

**Review:**

**Summary**
The paper considers the task of coarsely-disentangling a data-generating process into independent and shared modules. In more detail, it is assumed that each observation in a given dataset was generated by exactly one out of a set of independent generative processes referred to as independent causal mechanisms (ICM) in combination with a global mechanism that is shared across modules. The paper proposed the use of a mixture prior to model separate ICMs within a single architecture/generative model, chosen to be a GAN (Wasserstein-GAN with gradient penalty). Learning is performed by combining the GAN loss with self-supervision by using a separate encoder and adding loss terms resemblant of cycle-consistency-losses. The paper claims to prove a notion of identifiability of the coarse-grained modules in the sense of separation into disjoint manifolds and conducts some experiments on (variations of) MNIST and Fashion-MNIST, comparing their method against vanilla and disentangled VAE variants on some downstream tasks.

**Pros**
- the paper tackles a very interesting and highly-relevant topic (disentanglement with non-factorising latent space and connections to causality)
- the coarse-grained view and the separation into independent and shared mechanisms seems like a useful abstraction and (to the best of my knowledge) has not been investigated before in this form
- the authors support their proposed method with some theoretical analysis
- the latent traversal in Figure 2 and some of the quantitative results look promising

**Cons**
- even though I am very familiar with the topic of the submission, I found the paper very hard to follow
- many of the statements relating to causality and ICMs are inaccurate, misleading, or wrong (see detailed comments below)
- the notion of identifiability used in the paper is very unconventional and not consistent with prior work
- the assumed generative model is never fully specified (in particular, the mapping from latents to observations and the role of the mechanisms is not described in the text or given as formula) and the mixture prior is inconsistent with the graphical model given in Fig.1
- some of the cited works are misrepresented or wrongly protrayed (see detailed comments)


**Evaluation**
I think the paper tackles a very important problem and presents some interesting ideas. However, the paper contains too many errors, unclear parts, and inconsistencies in the current form which makes it very difficult to gain a clear picture of the proposed method and to assess its correctness. Significant improvements in terms of clarity and exposition are needed before publication and I therefore recommend rejection at the current stage.

**Detailed Comments and Questions**
- The ICM principle states that "the data generating process is composed of independent and autonomous modules that do not inform of influence each other" (Peters et al., 2017). I believe you should cite this. Note that this is different from the notion of isolated, as different ICMs may feed into each other. In fact this is how ICMs are conventionally understood in causality: each conditional distribution of a variable given its causal parents $P(X_i|PA_{X_i})$ is interpreted as an ICM, and together they form the data generating process for the joint $P(X_1, ..., X_n)=\prod_{i=1}^n P(X_i|PA_{X_i})$. ICMs are thus traditionally understood as independent modules that can be (re-)combined.
- I do not understand why the term "ICM-conditioned mechanism" is used: what is an "independent causal mechanism-conditioned mechanism"?
- Related Work: you mention three aspects but only two are given.
- Functional Causal Models: there are several key differences between FCMs and Bayesian networks (BN), so I think the presentation here is misguided. Firstly, BNs have no causal connotation to them but are only a way to represent a particular factorisation of a probability distribution. Causal graphical models or causal BNs on the other hand are endowed with a notion of intervention and are thus closer to FCMs. Perhaps this is what you intended to say? Moreover, it is unusual to refer to $P(X|PA_X)$ as a posterior---this is precisely what an ICM refers to and other common names include (causal) conditional or Markov kernel.
- the entire discussion of causality and FCMs misses the notion of intervention and counterfactual which is crucial for defining causal concepts
- The description of FCMs is incorrect: The deterministic relationships (structural equations) in SCMs are a set of assignments of the form $X_i:=f_i(PA_{X_i}, U_i)$ where $U_i$ are the unobserved variables and $f_i$ are indeed deterministic.
- I have never heard or read about this and would like to ask about the source of this characterisation: "If each function in FCM represents an autonomous mechanism, such FCM is called a structural model. Moreover, if each mechanism only determines the value of one and only one variable, such a structural model is called a structural causal model (SCM)." ? This seems strange to me.
- (Cai et al., 2019; Monti et al., 2020) are cited incorrectly: actually these aim to discover the graph which is different from learning an SCM as the former supports interventional reasoning, while the latter supports counterfactuals.
- Figure 1: the rectangles M_i are not described in the caption.
- ICA: the description of recent advances in ICA is incorrect: it refers to "requiring a relatively large number of independent components" when this should read "... large number of environments" or more generally values of the auxiliary variable which renders sources conditionally independent
- the specification of the mixture prior at the end of page 3 seems incorrect as it does not integrate to 1. Perhaps you meant to include mixture weights? However, this is also inconsistent with Fig.1: (c) would suggest a prior which factorises as $p(z)=p(z_C)p(z_S)\prod_{i=0}^N p(z_{M_i}|z_C)$, while (d) suggests $p(z)=p(z_S)\prod_{i=0}^N p(z_{M_i})$ both of which are different from the text. As I infer it from reading between the lines, $z_C$ is a categorical variable that switches between different mechanisms, and $z_M$, $z_S$ are both inputs to the generative process. Why the need for the many $z_M$'s? Would a simple mixture distribution on $z_M$ with weights and parameters depending on $z_C$ not do the job? If not, can you explain why?
- How is the isolation constraint implemented, i.e., how does it translate mathematically? Can you please specify the full generative process $p(z, x)$?
- the last paragraph in 3.2 is very hard to follow
- 3.3 suddenly mentions the encoder, but no encoder has been introduced up to this point
- can you justify the loss function in more detail?
- Wasserstain --> Wasserstein
- 4. The stated version of identifiability does not seem to make sense to me. It is almost always possible to learn the ground truth model, the question of identifiability is whether we are *guaranteed* to learn the right model. The cited work of Khemakem et al (2020) provides a nice and intuitive definition of identifiability in terms of equivalence classes in parameter space that seems more principled than the notion used here.
- Thm.3 This is the first time M_k is mentioned
- I am not sure what the point of the paragraph at the end of page 5 is? This does not really provide any intuition on the Theorem or its proof. Moreover, you assume $h:Z\rightarrow Z$ and then have $h(z)\not\in Z$ which seems to contradict each other. Please clarify.
- the paper refers to "co-variant shift" throughout; to the best of my knowledge "covariate shift" is the accepted terminology. is the different notation intentional, and if so can you clarify the difference?
- some experiments are not described in sufficient detail (e.g., what is being predicted in 5.2? why are there no error bars in Tables 1 or 3)
- the paper would benefit considerably from some professional proofreading and grammar checking (though this is not a key factor in my evaluation)



POST-REBUTTAL UPDATE:
I thank the authors for the detailed response. Based on the proposed changes I will slightly increase my score, but I still believe the paper needs additional work before meriting publication.

---

> ### Author Response · Authors · 2020-11-21
> **[Rebuttal 3/3] Identifiability result clarified; The proposed model is further specified; All the remainings are fixed**
>
> $\bullet\ $ The stated version of identifiability does not seem to make sense to me.
>
> $-$ Our identifiability result is consistent with previous disentangled representations literature [1, 2]. We first proved that the learned generative model g equals to the ground-truth g by up to a smooth-bijective transformation h. We have defined function-space equivalence relationship: $g \sim_h \tilde{g} \Leftrightarrow g^{-1}(\mathbf{x}) = h(\tilde{g}^{-1}(\tilde{\mathbf{x}})), \forall \mathbf{x} \in \mathcal{X}$ and identifiability: $p_{g}(\mathbf{x}) = p_{\tilde{g}}(\mathbf{x}) \Rightarrow g \sim_h \tilde{g}$. See Definition 1 and 2 in the revised paper for more details. We further proved that the transformation h does not entangle the mechanisms using Theorem 3.
>
> $\bullet\ $ Thm.3 This is the first time M_k is mentioned
>
> $-$ We have added the description of M_k in the introduction in the revised paper.
>
> $\bullet\ $ The point of the paragraph at the end of page 5 is unclear. The statements contradict each other.
>
> $-$ That paragraph describes the contradictory situations that we used in the proof. We have merged the paragraph to the proof in the revised paper.
>
> $-$ The contradiction is part of the proof. We prove the theorem by showing that if the generative model does not obey the desirable properties, the contradiction occurs. Thus, the generative model must obey the properties to avoid contradicting the assumption.
>
> $\bullet\ $ the paper refers to "co-variant shift" throughout, is it different from “covariant”?
>
> $-$ “co-variant” is the same as “covariant”, fixed in the revision.
>
> $\bullet\ $ some experiments are not described in sufficient detail
>
> $-$ We have improved the experimental description. In the second paragraph of section 5.2, we mentioned that we tested the prediction accuracy of GBT (the downstream model) on the test set. The prediction targets were the labels. We did not put error bars in Tables 1 and 3 because they were not describing the repeated experiments. Instead, the numbers in the table were the averages of multiple different repeated experiments. Specifically, we were averaging the average accuracy under different covariant shift strengths. The error bar was not applicable in this scenario. We reported the average variation across different experiments in Table 2.
>
> $\bullet\ $ the paper would benefit considerably from some professional proofreading and grammar checking
>
> $-$ We have made additional effort to improve the writing.
>
> **Reference:**
>
> [1] Locatello, Francesco, et al. "Challenging common assumptions in the unsupervised learning of disentangled representations." international conference on machine learning. PMLR, 2019.
>
> [2] Locatello, Francesco, et al. "Weakly-Supervised Disentanglement Without Compromises." arXiv preprint arXiv:2002.02886 (2020).
>
> [3] Khemakhem, Ilyes, et al. "Variational autoencoders and nonlinear ica: A unifying framework." International Conference on Artificial Intelligence and Statistics. 2020.
>
> [4] Pearl, Judea. "Causality: Models, Reasoning and Inference." Cambridge, UK: CambridgeUniversityPress (2000).
>
> [5] Bongers, Stephan, et al. "Foundations of structural causal models with cycles and latent variables." arXiv preprint arXiv:1611.06221 (2016).
>
> [6] Mooij, Joris, and Tom Heskes. "Cyclic causal discovery from continuous equilibrium data." arXiv preprint arXiv:1309.6849 (2013).
>
> [7] Mooij, Joris M., et al. "Distinguishing cause from effect using observational data: methods and benchmarks." The Journal of Machine Learning Research 17.1 (2016): 1103-1204.

---

> ### Author Response · Authors · 2020-11-21
> **[Rebuttal 2/3] Identifiability result clarified; The proposed model is further specified; All the remainings are fixed**
>
> $\bullet\ $ the specification of the mixture prior at the end of page 3 seems incorrect as it does not integrate to 1.
>
> $-$ Our prior is correctly designed but it does not follow the conventional Gaussian mixture distribution. We rewerite our prior as follow:
>
> $p(\mathbf{z}) = p(\mathbf{z}_{S}) \frac{\sum _{z_C} \prod _{k=1}^{N} p(\mathbf{z} _{M_k} \mid z_C)}{Z} = p(\mathbf{z}_S) \sum _{k=1}^{N} \frac{p_k(\mathbf{z}_M)}{Z}$
>
> $p(\mathbf{z} _{M_k} \mid z_C) = \mathcal{N}(\mathbf{0}, \mathbf{I}), z_C = k$
>
> $p(\mathbf{z} _{M_k} \mid z_C) = \mathbf{0} , \mathrm{otherwise.}$
>
> where $Z = \sum_{k=1}^{N} \int_{\mathbf{z}_M} p_k(\mathbf{z}_M) d\mathbf{z}_M$. We convert the sum of the joint conditional distribution to our unconventional mixture distribution by defining $p_1(\mathbf{z}_M) = [\mathcal{N}(\mathbf{0}, \mathbf{I}), \mathbf{0}, ..., \mathbf{0}]$, $p_2(\mathbf{z}_M) = [\mathbf{0}, \mathcal{N}(\mathbf{0}, \mathbf{I}), ..., \mathbf{0}]$ and so on. This change is available in the revised section 3.2
>
> $-$ The unconventional mixture prior is consistent with Figure 1(d) because for each distribution components, only one $\mathbf{z} _{M_k}$ is activated.
>
> $\bullet\ $ How is the isolation constraint implemented, i.e., how does it translate mathematically
>
> $-$ Assume we have $i, j \in \{1, …, N\}$. If we have $\mathbf{z}_{M_i} \neq \mathbf{0}$, we have $\mathbf{z}_{M_j} = \mathbf{0}, \forall j \neq i$. In colloquial words, only one $\mathbf{z}_{M_i}$ is activated at each time.
>
> $\bullet\ $ the last paragraph in 3.2 is very hard to follow
>
> $-$ In the last paragraph of section 3.2, we discussed the advantage of our unconventional mixture prior over the existing disentangled prior which is an isotropic Gaussian. To see this, consider that by definition, each $z_{i}$ in disentangled prior $\mathbf{z}$, which admits a density $p(\mathbf{z}) = \prod_{i=1}^{d} p(z_{i})$, represents one type of variations in disentangled representations. With the disentangled prior, we can generate data through the generative model by setting each $z_{i}$ to an arbitrary value. If we use each $z_i$ to represent each cause, the prior would allow the generative mechanism to generate data by arbitrarily combining the causes. This contradicts our assumption that each cluster in the data distribution has its distinct cause and none of the data is generated by the combined causes. To resolve this conflict, we need to either create a new dataset or adopt the unconventional mixture prior. The paragraph has been revised accordingly.
>
> $\bullet\ $ 3.3 suddenly mentions the encoder, but no encoder has been introduced up to this point.
>
> $-$ The encoder is only related to the self-supervision which is described in section 3.3. This encoder is a basic encoder that maps high dimensional data to low dimensional space. It is not the variational auto-encoder which computes a posterior. We want to disentangle the ICMs in the generative network instead of the encoder network.
>
> $\bullet\ $ can you justify the loss function in more detail?
>
> $-$ We have added the justification to section 3.3. Below is the justification:
>
> The loss function above can be decomposed into four parts: 1) $W(\mathbb{P} _{g} || \mathbb{P} _{r})$ is the WGAN loss. 2) $\mathbb{E} _{\mathbf{z} \sim P(\mathbf{z})} \mathcal{H}(\mathbf{z} _{C}, E(G(\mathbf{z})) _{C})$ is the self-supervision loss that enforces the disentanglement between generated data. 3) $\mathbb{E} _{\mathbf{x} \sim p(\mathbf{x})} ||\mathbf{x} - G(E(\mathbf{x}))|| _{2}^{2}$ is the forward cycle-consistent loss, which enforces $\mathbf{x} \approx G(E(\mathbf{x}))$. The encoder takes the generated data from the generator as input. However, the generated data distribution may differ from the real data distribution. Thus, certain data from the real distribution may become out-of-distribution (OOD) for the encoder. Such distribution divergence hurts the performance of downstream tasks. The forward cycle-consistency mitigates the distribution divergence by encouraging the ICM model to cover the whole real distribution. 4) $\mathbb{E} _{\mathbf{z} \sim p(\mathbf{z})} ||\mathbf{z} _{M} - E(G(\mathbf{z})) _{M}|| _{2}^{2} + \beta _{S}\mathbb{E} _{\mathbf{z} \sim p(\mathbf{z})} ||\mathbf{z} _{S} - E(G(\mathbf{z})) _{S}|| _{2}^{2}$ is the backward cycle-consistent loss, which enforces $\mathbf{z} \approx E(G(\mathbf{z}))$. The backward cycle-consistency prevents the generator from discarding certain causes in the generation process.

---

> ### Author Response · Authors · 2020-11-21
> **[Rebuttal 1/3] Identifiability result clarified; The proposed model is further specified; All the remainings are fixed**
>
> We thank the reviewer for the detailed suggestions and the very constructive feedback on our paper!
>
> **Response to cons:**
>
> $\bullet\ $ even though I am very familiar with the topic of the submission, I found the paper very hard to follow
>
> $-$ Our paper was written from a disentangled representations perspective. We have revised the paper to ease reading from a causality perspective.
>
> $\bullet\ $ many of the statements relating to causality and ICMs are inaccurate, misleading, or wrong
>
> $-$ We have added sources for descriptions that are strange to the reviewer (see detailed response below). We have fixed related issues in the revised paper.
>
> $\bullet\ $ the notion of identifiability used in the paper is very unconventional and not consistent with prior work
>
> $-$ The identifiability result is conventional in disentangled representations and is consistent with prior works that we cited [1, 2]. Identifying the disentangled model is different from identifying the right distribution parameters [3], which may be more familiar to the reviewer. We have bridged the gap between the two types of identifiability in the detailed response below and the revised section 4.
>
> $\bullet\ $ the assumed generative model is never fully specified
>
> $-$ We specified the generative model and the mixture prior in the original section 3. We have explained the relationship between the mechanisms and the generative model in the detailed response below. All the responses have also been integrated into the revised section 3.
>
> $\bullet\ $ some of the cited works are misrepresented or wrongly portrayed
>
> $-$ We have updated the cited works that are pointed out by the reviewer.
>
> **Response to detailed comments and questions:**
>
> $\bullet\ $ ICMs are not isolated. ICMs are traditionally understood as independent modules that can be (re-)combined
>
> $-$ We did not mention that the isolated mechanisms can not be (re)-combined. In fact, we assume that the data is generated by combining the generative mechanism and the shared mechanism. We have changed “isolate” to “disentangle” to avoid confusion.
>
> $\bullet\ $ I do not understand why the term "ICM-conditioned mechanism" is used
>
> $-$ We have renamed the “ICM-conditioned mechanism” to “generative mechanism”.
>
> $\bullet\ $ Related Work: you mention three aspects but only two are given.
>
> $-$ We have fixed the typo.
>
> $\bullet\ $ Functional Causal Models: there are several key differences between FCMs and Bayesian networks (BN). Moreover, it is unusual to refer to $P(X \mid PA_{X})$ as a posterior.
>
> $-$ We have changed the “posterior” to “conditional distribution”, and other issues have been fixed.
>
> $\bullet\ $ the entire discussion of causality and FCMs misses the notion of intervention and counterfactual which is crucial for defining causal concepts
>
> $-$ The intervention is an important concept but is not directly relevant to our work. This is because all the variables in our model are root nodes except for the observation variable $\mathbf{x}$. We mentioned that "As we only apply interventions on the root node, such interventions are equivalent to conditional generation" in the experiment section.
>
> $\bullet\ $ The description of FCMs is incorrect ....
>
> $-$ We believe we are describing the same concept. We mentioned that the uncertainty is introduced via the assumption that certain variables in the functions are not observed. If we manually extract the unobserved variables from $PA_{\mathbf{x}}$ and let them be $U_{\mathbf{x}}$, we have the equation that is mentioned by the reviewer.
>
> $\bullet\ $ I have never heard or read about this and would like to ask about the source of this characterisation: “...”.
>
> $-$ The source is Judea Pearl's causality book [4], section 1.4.
>
> $\bullet\ $ (Cai et al., 2019; Monti et al., 2020) are cited incorrectly.
>
> $-$ We have updated citations with [5, 6, 7].
>
> $\bullet\ $ Figure 1: the rectangles M_i are not described in the caption.
>
> $-$ Each rectangle represents a composition of the generative mechanism and the shared mechanism. We have added the description to the caption in the revised version.
>
> $\bullet\ $ ICA: the description of recent advances in ICA is incorrect: ...
>
> $-$ We have changed the description to “values of auxiliary variables”.

---

### Author Response · Authors · 2020-11-24
**Revision Summary**

We thank all the reviewers for their valuable comments. We have revised our paper accordingly. Please kindly refer to the updated version. Below is the revision summary:

- We have clarified the independent causal mechanisms principle in the first paragraph of section 1. We have also added examples of the causal mechanisms.

- We have specified the relationship between causal mechanisms and our generative model in the third paragraph of section 1.

- We have renamed our “mixture prior” as “unconventional mixture prior” in section 3.2. We have shown how to derive the unconventional mixture prior from a joint conditional distribution in the first paragraph of section 3.2.

- We have added the definitions of function-space equivalent relationship (Definition 1) and function-space identifiability (Definition 2) to section 4. We have discussed why the parameter-space identifiability from nonlinear ICA literature can not guarantee the disentanglement in the second paragraph of section 4.

- Other issues have also been fixed. The clarity has been improved.

---

### Decision · Program_Chairs · 2021-01-07
**Final Decision**

**Decision:**

Reject

**Comment:**

This paper proposes to learn representations in an unsupervised manner using a generative model in which observations are generated by combining independent causal mechanisms (ICMs), in combination with a global mechanism. The authors introduce an unconventional mixture prior for the shared and independent components of the representation and train an encoder, discriminator and generator using a Wasserstein GAN with additional terms that enforce consistency in the data and latent space. Experiments consider variations of MNIST and Fashion-MNIST and perform comparisons against a standard VAE, a β-VAE, and the Ada-GVAE.

Reviewers are broadly in agreement that this submission is not ready for publication in its current form. R4 in particular has left very detailed comments regarding clarity. The authors were able to in part address these comments, and R4 raised their score in response. That said, from a read of the manuscript in its latest form, the metareviewer (who is very familiar with literature on disentangled representations) is inclined to agree with the reviewers that this is work that has value, but is very difficult to follow in its current form. The metareviewer would like to suggest that the authors regroup, think carefully about how to improve clarity (in addition to addressing concrete points raised in reviews) and resubmit to a different venue.